# A Quantitative Analysis Model Established to Determine the Concentration of Each Source in Mixed Astaxanthin from Different Sources

**DOI:** 10.3390/molecules25030628

**Published:** 2020-01-31

**Authors:** Xiping Du, Xin Wang, Manli Bai, Shaosong Liu, Gaoling Huang, Qin Zhang, Hui Ni, Feng Chen

**Affiliations:** 1College of Food and Biological Engineering, Jimei University, Xiamen 361021, Fujian, China; xipingdu@jmu.edu.cn (X.D.); xinwang714@163.com (X.W.); manliact@163.com (M.B.); hgaol@jmu.edu.cn (G.H.); nihui@jmu.edu.cn (H.N.); 2Fujian Provincial Key Laboratory of Food Microbiology and Enzyme Engineering, Xiamen 361021, Fujian, China; 3Research Center of Food Biotechnology of Xiamen City, Xiamen 361021, Fujian, China; 4Key Laboratory of Systemic Utilization and In-depth Processing of Economic Seaweed, Xiamen Southern Ocean Technology Center of China, Xiamen 361021, Fujian, China; 5Plant Introduction & Quarantine and Plant Product Key Laboratory of Xiamen City, Xiamen Overseas Chinese Subtropical Plant Introduction Garden, Xiamen 361002, Fujian, China; lss0001888@163.com; 6Department of Food, Nutrition and Packaging Sciences, Clemson University, Clemson, SC 29634, USA; Fchen@Clemson.edu

**Keywords:** quantitative analysis model, stereoisomers, establishment, astaxanthin, verification

## Abstract

Astaxanthin from different sources possesses different biological activities and optical isomers. The ingredients of astaxanthin mixtures from different sources on the market have often been mislabeled. Therefore, it is important to determine the sources of astaxanthin and their respective concentrations in a mixture. To solve this problem, a quantitative analysis model was established and further verified. The results showed that the deviation between the calculated concentration and the actual concentration ranged from 0 to 7 µg/mL, and the recovery rate was between 88.90% and 103.56%. This indicates that the quantitative analysis model of astaxanthin was feasible and reliable. This study not only has important applications in the astaxanthin mixture component determination but may also shed light on the quantitative analysis of other sample mixtures with stereoisomers from different sources.

## 1. Introduction

Astaxanthin (3,3′-dihydroxy-β,β′-carotene-4,4′-dione), which is a ketocarotenoid oxidized from β-carotene [1,2], primarily comes from *Phaffia rhodozyma*, *Haematococcus pluvialis*, and chemical synthesis. Due to the presence of conjugated double bonds and α-hydroxyketones in its structure, astaxanthin has the strongest antioxidant activity among identified chemicals to date [3] and is called a ‘super antioxidant.’ The antioxidant capacity of astaxanthin is 500 times that of vitamin E and 10 times that of β-carotene [4,5]. Additionally, astaxanthin is the only carotenoid that can cross the blood brain barrier [6], which can protect the blood brain barrier from damage caused by oxidation. In addition, astaxanthin shows other important biological activities, including anti-obesity [7], anti-tumor [8], anti-inflammatory [9], anti-atherosclerotic [10], and immune-boosting properties [11], among others.

Astaxanthin exists as two enantiomers (*3R,3′R* and *3S,3′S*) in a meso form (*3S,3′R*) because it has two stereo-genic carbon atoms at the C3 and C3′ positions [12]. *P. rhodozyma* astaxanthin is mainly the (*3R,3′R*) isomer [13], *H. pluvialis* astaxanthin is mainly the (*3S,3′S*) isomer [14], and the chemically synthesized astaxanthin has a stereoisomeric ratio of 1:2:1 of the (*3S,3′S*), (*3S,3′R*), (*3R,3′R*) isomers [13,15]. Synthetic astaxanthin is widely used as an additive in fish feed, whereas naturally derived astaxanthin (mainly from *P. rhodozyma* and *H. pluvialis*) can be utilized in food and medicine [6,16]. Researchers have shown that the bioactivities of astaxanthin vary depending on its configuration [17], e.g., the potency of the chemical antioxidant activity is (*3S,3′S*)-astaxanthin > (*3R,3′R*)-astaxanthin > (*3S,3′R*)-astaxanthin [13,18]. In recent years, methods have been developed to distinguish astaxanthin stereoisomers. Three groups of rainbow trout were fed for 60 days of diets containing astaxanthin from three different sources. A characteristic distribution of astaxanthin stereoisomers was detected for each pigment source and such distribution were reproduced in the flesh of trout fed with that source [19]. Grewe et al. determined the configuration of astaxanthin derived from different microorganisms using a Chiralcel OD-RH column [20]. Subramanian et al. identified geometrical isomers of astaxanthin by Raman spectroscopy and UV–Vis absorption spectroscopy [21]. Wang et al. reported a direct and baseline separation method for all stereoisomers of all-trans-astaxanthin and other structurally-related carotenoids [15].The differences in the relative ratios of the configurational isomers of astaxanthin can be applied to distinguish aqua-cultured and wild salmon and identify astaxanthins derived from different sources [22]. A striking difference in the composition of astaxanthin optical isomers in *Chlamydomonas nivalis* was found to be concerned with geographically distinct regions with high-performance liquid chromatography (HPLC) methods [12]. However, former methods only studied the identification and qualitative analysis of the stereoisomers of a single source of astaxanthin. Astaxanthin from different sources possesses many optical isomers and each has different biological activities and economic value. In the market, the astaxanthins from different sources were possibly mixed as the color additive for salmon and incorrectly labeled in the content claims in pursuing profit ($2500 USD/kg, worldwide turnover of several hundred thousand kg per year [20]). It is required to identify the astaxanthin category from the mixture and quantify the astaxanthin of each source to justify the statements and claims, but identification of the sources and quantitative analysis of the concentration of each differently sourced astaxanthin in a mixture have not been reported.

In this study, we used HPLC data to establish a quantitative analysis model for the first time to determine the sources and concentrations of each astaxanthin component in a mixed sample. This study not only has important applications in the apportionment of astaxanthin mixtures but may also shed light on quantitative analysis of other sample mixtures with stereoisomers from different sources.

## 2. Results and Discussion

### 2.1. Identification of Astaxanthin from Different Sources Using High-Performance Liquid Chromatography (HPLC)

Three different astaxanthins, sourced from *P. rhodozyma*, *H. pluvialis*, and chemical synthesis, were analyzed by HPLC using a ZORBAX SB-C18 reversed-phase column (Figure 1A) and a chiral CHIRALPAK IC column (Figure 1B). The astaxanthins from different sources shared one absorbance peak at the same retention time of 13.667 min in the reversed-phase column, and no enantiomers could be separated (Figure 1A). The content of astaxanthin from different sources was almost the same. The three configurational isomers of astaxanthin could be separated in the CHIRALPAK IC column (Daicel Chiral Technologies Co., Ltd., Shanghai, China) (Figure 1B) in the eluting order of (*3S,3′S*), (*3S,3′R*), and (*3R,3′R*), which agrees with the results of Řezanka [12]. It can also be seen that astaxanthin from *P. rhodozyma* had configurations of (*3R,3’R*) and (*3S,3′R*).The astaxanthin from *H. pluvialis* showed a composition of the (*3S,3′S*) and (*3S,3′R*) enantiomers, and the synthetic astaxanthin contained (*3S,3′S*), (*3S,3′R*), and (*3R,3′R*) configurations. 

The peak areas of the enantiomers of astaxanthin from the three sources (Figure 1B) are shown in Table 1. *P. rhodozyma* astaxanthin was composed of dextral (*3R,3′R*) and racemic (*3S,3′R*) isomers at a ratio close to 15:1. *H. pluvialis* astaxanthin consisted of levo (*3S,3′S*) and racemic (*3S,3′R*) isomers at a ratio of 3:1. The synthetic astaxanthin contained the three levo (*3S,3′S*), racemic (*3S,3′R*) and dextral (*3R,3′R*) isomers at a ratio of 1:2:1. The configurations of the synthetic astaxanthin and their ratio were in agreement with those reported by Řezanka [12]. The astaxanthin from *P. rhodozyma* was observed as having one more enantiomer, the (*3S,3′R*) enantiomer, than the former report [13]. Furthermore, a majority of the astaxanthin from *H. pluvialis* was (*3S,3′S*), which is consistent with the results of Sun et al. [14], but small amounts of (*3S,3*′*R*)-astaxanthin was present in our separation.

The astaxanthin from one source possesses different isomers and one source of astaxanthin could be easily quantified by the standard curve established by the HPLC result, while the amount of each source astaxanthin from the mixture of different sources could not be directly identified from a standard curve due to part overlap of peaks between isomers from different sources.

### 2.2. Establishment of the Quantitative Analysis Model

Although individual sources of astaxanthin consisted of different isomers with constant isomeric ratios, it was difficult for us to analyze the composition of a mixture of several-source astaxanthin and determine its various sources. To distinguish astaxanthin from several different sources and quantify every category, we established a quantitative analysis model based on the experimental results of HPLC. The model was built by the following two principles. 

One principle was to establish the calibration curve of one stereoisomer of astaxanthin from a single source. The calibration curve reflects the relationship between the peak area of one stereoisomer and its concentration, which can be expressed as Equation (1).
(1)$$y_{mi}=a_{mi}x_{i}+b_{mi}$$
where *y_mi_* is the peak area of one enantiomer from one source (*m* = 1, 2, or 3, representing the number of enantiomers; *i* = 1, 2…*n*, representing the number of sources) and *x_i_* denotes the concentration of astaxanthin from one source.

The other principle was the additive property of the peak areas of an enantiomer, which was contributed to by all astaxanthin sources. The peak area of one stereoisomer in the mixed sample can be expressed as Equation (2).
(2)$$S_{m}=\overset{n}{\underset{i=1}{\sum }}y_{mi}$$

For embodiment, we assumed that the mixture was composed of two substances with each having three stereoisomers at different ratios. The quantitative analysis model can be described by combining Equation (1) and Equation (2).
(3)$$S_{1}=a_{11}x_{1}+b_{11}+a_{12}x_{2}+b_{12}$$
(4)$$S_{2}=a_{21}x_{1}+b_{21}+a_{22}x_{2}+b_{22}$$
(5)$$S_{3}=a_{31}x_{1}+b_{31}+a_{32}x_{2}+b_{32}$$
where the concentrations of substances from the two sources are x_1_ and x_2_, respectively.*S_1_*, *S_2_*, and *S_3_* represent the total peak areas of the three enantiomers, respectively, and *a_mi_* and *b_mi_* are the slope and intercept of the calibration curves of *y_mi_*, respectively, for each enantiomer (*m* = 1, 2, 3, representing the number of enantiomers and *i* = 1, 2, representing the number of sources).

Similarly, for the mixture composed of three different sources of astaxanthin with three stereoisomers at different ratios, the quantitative analysis model can be expressed as follows.
(6)$$S_{1}=a_{11}x_{1}+b_{11}+a_{12}x_{2}+b_{12}+a_{13}x_{3}+b_{13}$$
(7)$$S_{2}=a_{21}x_{2}+b_{21}+a_{22}x_{2}+b_{22}+a_{23}x_{3}+b_{23}$$
(8)$$S_{3}=a_{31}x_{1}+b_{31}+a_{32}x_{2}+b_{32}+a_{33}x_{3}+b_{33}$$
where the concentrations of isomers from the three sources are x_1_, x_2_, and x_3_, respectively, and *S_1_*, *S_2_,* and *S_3_* represent the total peak areas of the three enantiomers in the mixed sample, respectively. The parameters *a_mi_* and *b_mi_* are the slope and intercept, respectively, which were obtained from the calibration curves of *y_mi_* (*m* = 1, 2, 3, representing the number of enantiomers and *i* = 1, 2, 3, representing the number of sources).

Analogously, a mixture composed of n different sources of astaxanthin with three stereoisomers at different ratios could be applied to the analytical model with the following equations.
(9)$$S_{1}=a_{11}x_{1}+a_{12}x_{2}+\ldots +a_{1n}x_{n}+k_{1}$$
(10)$$S_{2}=a_{21}x_{1}+a_{22}x_{2}+\ldots +a_{2n}x_{n}+k_{2}$$
(11)$$S_{3}=a_{31}x_{1}+a_{32}x_{2}+\ldots +a_{3n}x_{n}+k_{3}$$
where $k_{1}=b_{11}+b_{12}+\ldots +b_{1n}$; $k_{2}=b_{21}+b_{22}+\ldots +b_{2n}$; $k_{3}=b_{31}+b_{32}+\ldots +b_{3n}$. The concentrations of substances from several different sources are *x_1_*, *x_2_* … *x_n_*, and *S_1_*, *S_2_,* and *S_3_* represent the total peak areas of the three enantiomers, respectively. The parameters *a_mi_* and *b_mi_* are the slope and intercept of the calibration curve of *y_mi_* (*m* = 1, 2, 3, representing the number of enantiomers and *i* = 1, 2…*n*, representing the number of sources). It should be mentioned that assuming three stereoisomers in all cases was mainly because only three stereoisomers at most could coexist in one source.

It can be seen from Equations (9)–(11) that the total peak area of the same stereoisomer from different sources of the same substance in a mixture and the calibration curves of different stereoisomers of each source were known. From this information, the sources and their respective concentrations in the mixture could be quantified.

Therefore, the model could be utilized to calculate the amount of each astaxanthin based on the experimental result of HPLC. This model could be expected to identify the astaxanthin category from the mixture of different sources and quantify the astaxanthin of each source, which can resolve the complicated quantitative problem due to part overlap among the isomers from different sources.

### 2.3. Detection of Astaxanthin with the Quantitative Analysis Model 

Figure 2 shows the HPLC-separated result of *P. rhodozyma*, *H. pluvialis*, synthetic astaxanthin, and their mixture. The peaks of the mixture had retention times of 10.23 min, 11.81 min, and 13.64 min, which were the same as the three enantiomers. However, we cannot qualify the composition of the mixture as judged by the retention time since the enantiomers in a single-source, two-source, or three-source astaxanthin mixture will produce the same retention times. It should be noted that the intensity ratio between isomers of single-source astaxanthin was constant, which include the dextral (*3R,3′R*) and racemic (*3S,3′R*) isomers at an intensity ratio of 15:1 in *P. rhodozyma*, the levo (*3S,3*′*S*) and racemic (*3S,3′R*) isomers at a ratio of 3:1 in *H. pluvialis*, and the levo (*3S,3′S*), racemic (*3S,3′R*) and dextral (*3R,3′R*) isomers at a ratio of 1:2:1 in synthetic astaxanthin. However, when two-source or three-source astaxanthin mixtures with different concentrations were analyzed (Figure 2), it was difficult for us to not only distinguish the astaxanthin from different sources through their relative signal ratios but also to directly quantify the isomers from each source.

However, according to the two steps of the mathematical model in the previous section, we can establish a quantitative analysis model for astaxanthin, from which the astaxanthin mixtures can be quantitatively analyzed, the composition can be determined, and the concentration of each source of astaxanthin can be achieved.

First, the calibration curves of (*3R,3′R*)- and (*3S,3′R*)-astaxanthin from *P. rhodozyma*, (*3S,3′S*)- and (*3S,3′R*)-astaxanthin from *H. pluvialis*, and of (*3S,3′S*), (*3S,3′R*) and (*3R,3′R*) from synthetic astaxanthin were established with the concentrations and peak areas, as shown in Appendix A. The linear relationships of all astaxanthin stereoisomers from the three sources were fairly good, and the correlation coefficients were all above 0.999. The slope a and intercept b of the calibration curves were obtained (Table 2), which were the required parameters for the model.

Second, the quantitative analysis model for astaxanthin determination could be implemented when the parameters of a, b, and k in the model described as Equations (9), (10), and (11) were defined. The parameter k can be determined as the sum of all b values. These parameters could be derived from the previously mentioned standard curves, as shown in Table 2. Then, the parameters a, b, and k can be inputted in the quantitative analysis model (Equations (9)–(11)), and the quantitative analysis of astaxanthin can be achieved.
(12)$$S_{1}=26.12x_{2}+16.595x_{3}-4.0403$$
(13)$$S_{2}=2.3151x_{1}+9.4757x_{2}+34.198x_{3}-2.3492$$
(14)$$S_{3}=32.543x_{1}+17.259x_{3}-10.6925$$
where *x_1_*, *x_2_*, and *x_3_* represent the concentrations of *P. rhodozyma*, *H. pluvialis*, and synthetic astaxanthin, respectively.*S_1_*, *S_2_*, and *S_3_* represent the total peak areas of (*3S,3′S*)-, (*3S,3′R*)-, and (*3R,3′R*)-astaxanthin, as shown in the HPLC spectra.

Lastly, the sources of astaxanthin and their concentrations could be determined when the peak areas obtained experimentally were put into the quantitative model (Equations (12), (13), and (14)), and the three-dimensional first-order equations of the model were solved.

This model was suitable for the quantitative analysis of a mixture of astaxanthin from several different sources with a complex sample due to several isomers being present in each source. Related reports in the literature were all aimed at the quantitative analysis of astaxanthin from one source. For example, Turujman et al. used liquid chromatography to quantify synthetic astaxanthin, which is a pigmentation additive in salmon [22], based on the difference in the relative proportions of astaxanthin isomers. Dissing et al. established a partial least square regression (PLSR) model to predict the concentration of astaxanthin in rainbow trout slices [23].

### 2.4. Verification of the Quantitative Analysis Model

When the mixture of *P. rhodozyma*, *H. pluvialis*, and synthetic astaxanthin were separated by chromatography, the different isomers were separated, and the identical isomers from each source overlapped, which leads to the overall peak areas of each stereoisomer, with *S_1_*, *S_2_*, and *S_3_* denoting (*3S,3′S*)-, (*3S,3′R*)-, and (*3R,3′R*)-astaxanthin, respectively. The concentration of each astaxanthin source could be obtained by putting the value of the above three peak areas into the model for solving the three-dimensional Equations (12), (13), and (14). The recovery rate, namely, the calculated concentration compared with the amount of astaxanthin added from each source for the experiments, was analyzed to verify the accuracy of the above model.

Table 3 shows the different concentration ratios of *P. rhodozyma*, *H. pluvialis*, and synthetic astaxanthin, the peak areas *S_1_*, *S_2_*, and *S_3_* of various isomers obtained from the HPLC spectra, and the calculated concentrations of three astaxanthins, *x_1_*, *x_2_*, and *x_3_*, by using the quantitative models. Figure 3 shows the recovery rate obtained by the model calculation when the three sources of astaxanthin were mixed in different proportions. When the concentration ratio of *P. rhodozyma*, *H. pluvialis*, and synthetic astaxanthin was 1:1:1, namely, the concentration of astaxanthin from all three sources was 25 µg/mL, the astaxanthin concentrations from each source were calculated to be 23.56, 24.92, and 26.35 µg/mL, respectively, according to the quantitative analysis model. The recovery rate was 94.2%. When the ratio was 1.0:0.2:0.2, namely, the three astaxanthin sources were at 25, 5, and 5 µg/mL, respectively, the concentrations of astaxanthin from the three sources were calculated to be 24.15, 5.23, and 4.96 µg/mL, respectively, by using the quantitative analysis model, and the recovery rate was 96.6%. When the concentration ratio was 1.0:1.5:0.5, namely, the three sources of astaxanthin were at 50, 75, and 25 µg/mL, respectively, the concentrations of astaxanthin from the three sources were determined to be 51.78, 74.55, and 29.27 µg/mL, respectively, by using the model, and the recovery rate reached 103.6%. When the concentration ratio of astaxanthin from the three sources was 1:3:4, i.e., at concentrations of 25, 75, and 100 µg/mL, respectively, the calculated concentrations of astaxanthin were 25.53, 75.03, and 105.52 µg/mL, respectively. The recovery rate was 102.1%.

When the astaxanthin from the three sources was mixed at different ratios, as shown in Table 3, the deviation between the calculated concentration and the actual concentration ranged from 0 to 7 µg/mL, and the recovery rate was between 88.90% and 103.56% (Figure 3). This indicated that the quantitative analysis model of astaxanthin was feasible and reliable, which could be utilized to identify the astaxanthin category and quantify the astaxanthin of each source from different sources and resolve the complicated quantitative problem of the isomers from different sources.

This model can not only be applied to the quantitative analysis of astaxanthin but also provide a new pathway for the quantitative analysis of mixtures of other natural products from different sources with each source having different stereoisomer ratios. It should also be noted that the peak areas of astaxanthin in the quantitative analysis model were related to the precision of the method. The analytical range of astaxanthin in the chromatographic column was 5 to 100 µg/mL, so the analytical range of the quantitative model was 5 to 100 µg/mL.

## 3. Experimental

### 3.1. Materials and Reagents 

Synthetic astaxanthin (USP, 100% pure) was purchased from Sigma-Aldrich (St. Louis, MO, USA).Astaxanthin from *P. rhodozyma* and *H. pluvialis* were obtained from our laboratory with purities of 100% and 95.6% (calculated by peak area normalization) [24,25], respectively. They were all stored at −20 °C. Hexane, acetonitrile, *tert*-butyl-methyl-ether, 2-propanol, and methanol were of a chromatographic grade and were purchased from Sigma-Aldrich. Dichloromethane was of an analytical grade (Sinopharm Chemical Reagent Co., Ltd., Shanghai, China).

### 3.2. Instrumentation

Analysis was carried out using an Agilent 1260 infinity quaternary liquid chromatograph (Agilent Technologies Deutschland GmbH, Böblingen, Germany) equipped with an auto injector, an online vacuum degasser, a G1311B/C quaternary pump, a G1329B autosampler, a G1330B thermostat, a G1316B thermostatted column compartment, and a G1314F variable wavelength detector (VWD). The columns applied in this work were a CHIRALPAK IC column (250 mm × 4.6 mm, I.D., 5 µm, Daicel Chiral Technologies Co., Ltd., Shanghai, China) and a ZORBAX SB-C18column (100 mm × 2.1 mm, I.D., 3.5 µm, Agilent Technologies, USA).

### 3.3. HPLC Analysis

Identification of the three different sources of astaxanthin were analyzed by HPLC with solvent system I consisting of ultrapure water (mobile phase A) and methanol (mobile phase B). The separation was performed on a ZORBAX SB-C18 column at a flow rate of 0.2 mL/min. The column temperature was 35 °C, the injection volume was 5.0 μL, and the detection wavelength was 474 nm. Gradient elution conditions are shown as follows: 85% to 100% mobile phase B in mobile phase A in the first 40 min, 100% mobile phase B from 40 to 58 min, and100% to 85% mobile phase B in mobile phase A from 58 to 66 min.

The three different sources of configurational isomers of astaxanthin were analyzed by HPLC with solvent system II consisting of acetonitrile (mobile phase C) and *tert*-butyl-methyl-ether (mobile phase D). Isocratic elution conditions were 0–25 min, 60% C and 40% D. The separation was performed on a CHIRALPAK IC column at a flow rate of 1 mL/min. The column temperature was 25 °C, the injection volume was 10 µL, and the detection wavelength was 476 nm.

### 3.4. Sample Preparation

Stock solutions of 300 μg/mL astaxanthin were prepared by dissolving 0.003 g three-source astaxanthin in acetonitrile and were immediately diluted to 225, 150, 90, 75, 45, 30, and 15 µg/mL prior to use. The three-source astaxanthin sample solutions were mixed at a certain ratio to obtain the mixture concentration shown in Table 3. Then, the mixture was filtered through a 0.22-µm microporous membrane. Lastly, the mixtures were measured by HPLC analysis in triplicate.

### 3.5. Statistical Analysis

All analyses were performed and evaluated in triplicate. Statistical analyses were performed using Microsoft Office Excel 2013. The results are expressed as the mean ± SD.

## 4. Conclusions

This study successfully established a quantitative analysis model that could be used to determine the sources and concentrations of each source of astaxanthin in mixed astaxanthin samples. The quantitative analysis model was verified, and the results showed that the deviation between the calculated concentration and the actual concentration ranged from 0 to 7 µg/mL. The recovery rate was between 88.90% and 103.56%. This indicated that the quantitative analysis model of astaxanthin was feasible and reliable. This study cannot only be applied in the analysis of mixed astaxanthin but also may castlight on quantitative analysis of other sample mixtures with stereoisomers from different sources.

## Figures and Tables

**Figure 1 molecules-25-00628-f001:**
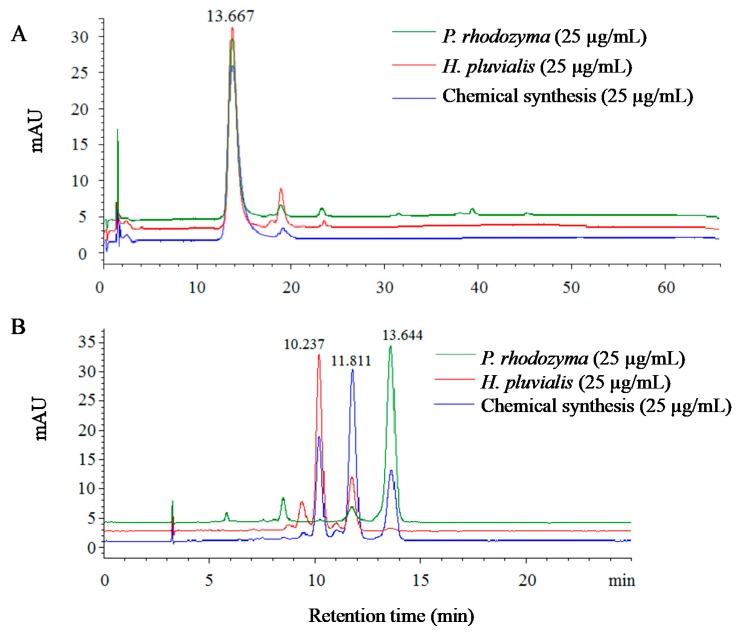
Separation of astaxanthin from *P. rhodozyma*, *H. pluvialis*, and chemical synthesis by high-performance liquid chromatography (HPLC) with ZORBAX SB-C18 (**A**) and CHIRALPAK IC (**B**) columns.

**Figure 2 molecules-25-00628-f002:**
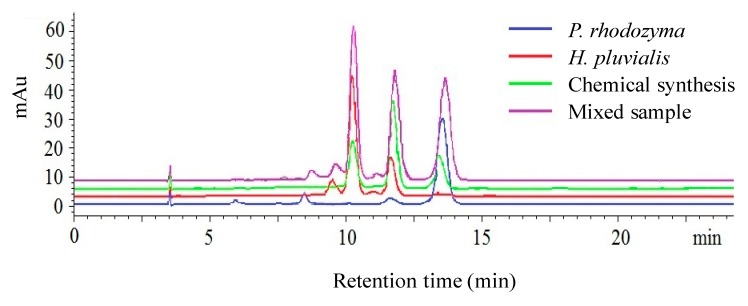
High-performance liquid chromatography (HPLC) analysis of *P. rhodozyma*, *H. pluvialis*, and synthetic astaxanthin at 25, 30, and 15 μg/mL, respectively, and of their mixture.

**Figure 3 molecules-25-00628-f003:**
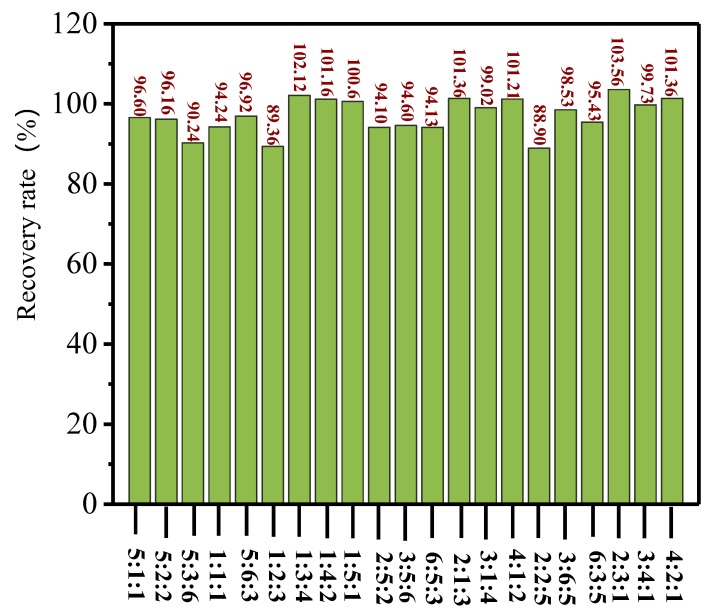
The recovery rate of astaxanthin from *P. rhodozyma* at different concentration ratios.

**Table 1 molecules-25-00628-t001:** Peak area of the individual enantiomers of astaxanthin sourced from *P. rhodozyma*, *H. pluvialis*, and chemical synthesis.

Isomer	*P. rhodozyma* (mAU*S)	*H. pluvialis* (mAU*S)	Chemical Synthesis (mAU*S)
*3S,3′S*	--	640.53	405.19
*3S,3′R*	55.87	222.55	828.27
*3R,3′R*	789.90	--	415.65

mAU*S represents peak area unit.

**Table 2 molecules-25-00628-t002:** Quantitative analysis model parameters for astaxanthin determination from *P. rhodozyma*, *H. pluvialis*,and chemical synthesis.

S	Calibration Curve	a	b	k
S_1_ = y_11_ + y_12_	y_11_ = 26.12x_2_ − 5.6712	26.12	−5.6712	k_1_ = −4.0403
	y_12_ = 16.595x_3_ + 1.6309	16.595	1.6309	
S_2_ = y_21_ + y_22_ + y_23_	y_21_ = 2.3151x_1_ − 0.4045	2.3151	−0.4045	k_2_ = −2.3492
	y_22_ = 9.4757x_2_ − 1.489	9.4757	−1.489	
	y_23_ = 34.198x_3_ − 0.4557	34.198	−0.4557	
S_3_ = y_31_ + y_32_	y_31_ = 32.543x_1_ − 7.7618	32.543	−7.7618	k_3_ = −10.6925
	y_32_ = 17.259x_3_ − 2.9307	17.259	−2.9307	

**Table 3 molecules-25-00628-t003:** Calculation and verification of the quantitative analysis model of mixed astaxanthin.

Actual Concentration (µg/mL)	S_1_ (mAU*S)	S_2_ (mAU*S)	S_3_ (mAU*S)	Calculated Concentration (µg/mL)	Recovery Rate (%)
25*x_1_* + 5*x_2_* + 5*x_3_*	215.80 ± 11.16	272.68 ± 9.21	861.04 ± 69.41	24.15*x_1_* + 5.23*x_2_* + 4.96*x_3_*	96.6
25*x_1_* + 25*x_2_* + 25*x_3_*	1084.81 ± 12.72	1189.62 ± 32.78	1211.06 ± 41.44	23.56*x_1_* + 24.92*x_2_* + 26.35*x_3_*	94.2
25*x_1_* + 75 *x_2_* + 100*x_3_*	3706.90 ± 111.08	4376.76 ± 11.01	2641.15 ± 110.07	25.53*x_1_* + 75.03*x_2_* + 105.52*x_3_*	102.1
10*x_1_* + 25*x_2_* + 10*x_3_*	804.46 ± 16.58	601.49 ± 15.88	472.68 ± 15.82	9.41*x_1_* + 24.41*x_2_* + 10.25*x_3_*	94.1
30*x_1_* +25 *x_2_* + 15*x_3_*	879.53 ± 15.09	818.29 ± 15.26	1174.93 ± 43.63	28.24*x_1_* + 23.99*x_2_* + 15.44*x_3_*	94.1
50*x_1_* + 75*x_2_* + 25*x_3_*	2429.42 ± 49.13	1825.07 ± 7.79	2179.89 ± 19.12	51.78*x_1_* + 74.55*x_2_* + 29.27*x_3_*	103.6
100*x_1_* +50*x_2_* + 25*x_3_*	1846.67 ± 29.74	1787.07 ± 5.19	3829.39 ± 18.43	101.36*x_1_* +50.91*x_2_* + 31.36*x_3_*	101.4

mAU*S represents the peak area unit.

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
