# Peer review of "A Quantitative Analysis Model Established to Determine the Concentration of Each Source in Mixed Astaxanthin from Different Sources"

_molecules, 2020, doi:10.3390/molecules25030628_

Round 1

Reviewer 1 Report

The Author reported in a very easy way a very gravel problems about Astaxanthin from different sources that by 19 optical isomers possess different biological activities.  I feel that the authors should be invited to revisit the manuscript with a view to increasing  results and discussions relationships before re-submitting. My suggestion would be to concentrate their attention on the investigated system presenting the result of each mixture. The study has important applications in the Astaxanthin mixtures determination but in order to be a valid point for analytical quantitative validation as to be presented as quantitative study and not  like a “a general quantitative “ as the author define their quantification model. That is the major concern about the paper. From a scientific point of view they have to evaluate separately the results in light of the analytical results without any other feeling. Why did they have to established a general model, this exception is not part of scientific knowledge. Moreover the study is well structured and the author, by using the presented datas they may approach the analytical problem without any other exception. Several part of the presented paper like chromatograms or other calibration curves may be part of supplementary additional material of the presented study.

Author Response

Response to Reviewer 1 Comments

 The Author reported in a very easy way a very gravel problems about Astaxanthin from different sources that by 19 optical isomers possess different biological activities.  I feel that the authors should be invited to revisit the manuscript with a view to increasing results and discussions relationships before re-submitting. My suggestion would be to concentrate their attention on the investigated system presenting the result of each mixture. The study has important applications in the Astaxanthin mixtures determination but in order to be a valid point for analytical quantitative validation as to be presented as quantitative study and not  like a “a general quantitative “ as the author define their quantification model. That is the major concern about the paper. From a scientific point of view they have to evaluate separately the results in light of the analytical results without any other feeling. Why did they have to established a general model, this exception is not part of scientific knowledge. Moreover the study is well structured and the author, by using the presented datas they may approach the analytical problem without any other exception. Several part of the presented paper like chromatograms or other calibration curves may be part of supplementary additional material of the presented study.

Response 1: according to reviewer’s view, we have added more discussion in subsetion 2.1, 2.2, 2.4 as follows.

2.1 The astaxanthin from one source possesses different isomers and one source of astaxanthin could be easily quantified by the standard curve established by the HPLC result, while the amount of each source astaxanthin from the mixture of different sources could not be directly identified from standard curve due to part overlap of peaks between isomers from different sources.

2.2 Therefore, the model could be utilized to calculate the amount of each astaxanthin based on the experimental result of HPLC. This model could be expected to identify the astaxanthin category from the mixture of different sources and quantify the astaxanthin of each source, which can resolve the complicated quantitative problem due to part overlap among the isomers from different sources.

2.4 This indicated that the quantitative analysis model of astaxanthin was feasible and reliable, which could be utilized to identify the astaxanthin category and quantify the astaxanthin of each source from different sources and resolve the complicated quantitative problem of the isomers from different sources.

Response 2: We agree with the reviewer’s view and replace “ a general quantitative model” with “a quantitative model”. All concerned with “a general model” in the context has been modified to “a model” or “an astaxanthin model” as followed.

Line 21 “a general quantitative analysis model” has been modified to “a quantitative analysis model”. Line 80 “a general quantitative analysis model“ has been modified to “a quantitative analysis model”. Line 121 “we established a generalized quantitative analysis model“ has been changed to “we established a quantitative analysis model”. Line 205 “when the parameters of a, b and k in the general model described as equations 9, 10, and 11 were defined” has been changed to “when the parameters of a, b and k in the model described as equations 9, 10, and 11 were defined”. Line 260 “This model can be applied not only to the quantitative analysis of astaxanthin but also to the quantitative analysis of mixtures of other natural products” has been modified to “This model can not only be applied to the quantitative analysis of astaxanthin but also provide a new pathway for the quantitative analysis of mixtures of other natural products”. Line 312 “This study successfully established a general quantitative analysis model” has been modified to “This study successfully established a quantitative analysis model”.

Response 3: according to reviewer’s view, we have put calibration curves in the part of supplementary additional material of the presented study.

Moderate English changes required.

Response: We tried our best to write sentences correctly in English. To avoid some literal and grammer mistakes, we asked some native speaker of English to verify whether the expression is correct in our manuscript.

Reviewer 2 Report

This document presents interesting information about a quantitative analysis model established to determine the concentration of astaxanthin from different sources. However, some modifications must be done to be considered for publication at Molecules.

Delete Table 4, this information can be included as text

Lines 261 – 271. Revise all subsection 3.4. Sample preparation and correct

Separate subsection 3.5. Statistical analysis and Section 4 (Conclusion).

Lines 87-89: Why these three isomers are presented in such sample? Explain and include in the text

More discussion is needed in subsections 2.2, 2.3 and 2.4 in the Results section. This, in order to putting out the relevance and originality of this investigation. 

Author Response

Response to Reviewer 2 Comments

Delete Table 4, this information can be included as text

Response: Table 4 has been deleted and this information has been included as following text.

Gradient elution conditions are shown as follows: 85% to 100% mobile phase B in mobile phase A in the first 40 min, 100% mobile phase B from 40 to 58 min, 100% to 85% mobile phase B in mobile phase A from 58 to 66 min.

Lines 261-271. Revise all subsection 3.4. Sample preparation and correct

Response: All subsection 3.4. Sample preparation has been revised and corrected as follows.

Stock solutions of 300 µg/mL astaxanthin were prepared by dissolving 0.003 g three-source astaxanthin in acetonitrile and were immediately diluted to 225, 150, 90, 75, 45, 30 and 15 µg/mL prior to use. The three-source astaxanthin sample solutions were mixed at a certain ratio to obtain the mixture concentration shown in Table 3. Then, the mixture was filtered through a 0.22 µm microporous membrane. Finally, the mixtures were measured by high-performance liquid chromatography (HPLC) analysis in triplicate.

Separate subsection 3.5. Statistical analysis and Section 4 (Conclusion).

Response: Subsection 3.5. Statistical analysis and Section 4 (Conclusion) has been separated.

Lines 87-89: Why these three isomers are presented in such sample? Explain and include in the text.

Response: according to the reviewer’s view, we have explained the reason why those isomers are presented in the sample and included in introduction.

Astaxanthin from different sources possesses many optical isomers and each has different biological activities and economic value. In the market, the astaxanthins from different sources were possibly be mixed as the color additive for salmon and incorrectly labeled in the content claims in pursuing profit (US$ 2500/kg, world wide turnover of several hundred thousand kg per year). It’s required to identify the astaxanthin category from the mixture and quantify the astaxanthin of each source to justify the statements and claims, ,but identification of the sources and quantitative analysis of the concentration of each differently sourced astaxanthin in a mixture have not been reported.

More discussion is needed in subsections 2.2, 2.3 and 2.4 in the Results section. This, in order to putting out the relevance and originality of this investigation.

Response: according to reviewer’s view, we have added more discussion in subsetion 2.1-2.4 as follows.

2.1 The astaxanthin from one source possesses different isomers and one source of astaxanthin could be easily quantified by the standard curve established by the HPLC result, while the amount of each source astaxanthin from the mixture of different sources could not be directly identified from standard curve due to part overlap of peaks between isomers from different sources.

2.2 Therefore, the model could be utilized to calculate the amount of each astaxanthin based on the experimental result of HPLC. This model could be expected to identify the astaxanthin category from the mixture of different sources and quantify the astaxanthin of each source, which can resolve the complicated quantitative problem due to part overlap among the isomers from different sources.

2.3 These parameters could be derived from the above mentioned standard curves as shown in Table 2. Then, the parameters a, b and k can be inputted in the quantitative analysis model (equation 9, 10, 11), and the quantitative analysis of astaxanthin can be achieved.

2.4 This indicated that the quantitative analysis model of astaxanthin was feasible and reliable, which could be utilized to identify the astaxanthin category and quantify the astaxanthin of each source from different sources and resolve the complicated quantitative problem of the isomers from different sources.

Reviewer 3 Report

Manuscript ID: molecules-696097

Recommendation: Accept after major changes

The authors in the manuscript “A Quantitative Analysis Model Established to Determine the Concentration of Each Source in Mixed Astaxanthin from Different Sources designated the aim to enable the components determination in astaxanthin mixture, at the same time pointing out the possibility of the quantitative analysis of other sample mixtures with stereoisomers from different sources.

The comments:

Line 18, Abstract, the first sentence should be corrected

Authors should provide the full Latin name of all mentioned species in the manuscript (Latin name, author’s name, family name). Afterwards, throughout the text the shorten forms should be used

Line38, instead of “Astaxanthin”, should be “astaxanthin”

Introduction should be expanded adding the valid data regarding the advantages and shortcomings of the used methods in the cited literature, and the novelty the applied method in the manuscript possessed.

Tables 1,3 - please give the explanation of the used abbreviations (mAU*S) in the legend of tables

Line 269, please clarify Table 4 mentioning, as in the text Table 4 contained the data regarding HPLC analysis

Please, give the comparison between the content of astaxanthin determined using ZORBAX SB-C18 column, where the separations of stereoisomers had not been achieved to the results obtained using CHIRALPAK IC column.

Prior to acceptance, please, perform the experiment with real samples, for example, determine the stereoisomeres in sample originated from salmon for which the color additive astaxanthin was used, in order to verify the suggested method..

Overall impression is that the authors successfully employed the mathematical model in solving the problem of quantitative and qualitative analysis of the stereoisomers.

Author Response

Response to Reviewer 3 Comments

Line 18, Abstract, the first sentence should be corrected

Response: The first sentences in abstracts “Astaxanthin comes from different sources that possess different biological activities and optical isomers”, has been crorrected to “Astaxanthin from different sources possess different biological activities and optical isomers”.

Authors should provide the full Latin name of all mentioned species in the manuscript (Latin name, author’s name, family name). Afterwards, throughout the text the shorten forms should be used

Response: The full Latin names of all mentioned species (Phaffia rhodozyma (Line 32), Haematococcus pluvialis (Line 32), Chlamydomonas nivalis (Line 62)) in the manuscript have provided. Afterwards, the shorten forms of them (P. rhodozyma and H. pluvialis) were used.

Line38, instead of “Astaxanthin”, should be “astaxanthin”

Response: “Astaxanthin” has been changed to “astaxanthin”.

Introduction should be expanded adding the valid data regarding the advantages and shortcomings of the used methods in the cited literature, and the novelty the applied method in the manuscript possessed.

Response: According to reviewer’s suggestions, we revised some contents in introduction as follows:

In recent years, methods have been developed to distinguish astaxanthin stereoisomers. Three groups of rainbow trout were fed for 60 days diets containing astaxanthin from three different sources. A characteristic distribution of astaxanthin stereoisomers was detected for each pigment sources and such distribution was reproduced in the flesh of trout fed with that source [19]. Grewe et al. determined the configuration of astaxanthin derived from different microorganisms using a Chiralcel OD-RH column [20]. Subramanian et al. identified geometrical isomers of astaxanthin by Raman spectroscopy and UV–Vis absorption spectroscopy [21]. Wang et al. reported a direct and baseline separation method for all stereoisomers of all-trans-astaxanthin and other structurally related carotenoids [15]. The differences in the relative ratios of the configurational isomers of astaxanthin  can be applied to distinguish aquacultured and wild salmon and identify astaxanthins derived from different sources [22]. A striking difference in the composition of astaxanthin optical isomers in Chlamydomonas nivalis was found to be concerned with geographically distinct regions with high-performance liquid chromatography (HPLC) methods However, former methods only studied the identification and qualitative analysis of the stereoisomers of a single source of astaxanthin. Astaxanthin from different sources possesses many optical isomers and each has different biological activities and economic value. In the market, the astaxanthins from different sources were possibly be mixed as the color additive for salmon and incorrectly labeled in the content claims in pursuing profit (US$ 2500/kg, world wide turnover of several hundred thousand kg per year [20]). It’s required to identify the astaxanthin category from the mixture and quantify the astaxanthin of each source to justify the statements and claims, but identification of the sources and quantitative analysis of the concentration of each differently sourced astaxanthin in a mixture have not been reported.

In this study, we used HPLC data to establish a quantitative analysis model for the first time to determine the sources and concentrations of each astaxanthin component in a mixed sample.

Tables 1,3 - please give the explanation of the used abbreviations (mAU*S) in the legend of tables

Response: the explanation of the used abbreviations (mAU*S) has been added in the legend of tables.

Line 269, please clarify Table 4 mentioning, as in the text Table 4 contained the data regarding HPLC analysis

Response: Thank you for your valuable comment. Table 4 in Line 269 should be revised to Table 3. Because the order of Molecules (1. Introduction, 2. Results and Discussion, 3. Experimental) is different from our original manuscript (1. Introduction, 2. Experimental, 3. Results and Discussion), We are sorry that we forgot to revise Table 4 in the text to Table 3 in the text, and now it has been revised.

Please, give the comparison between the content of astaxanthin determined using ZORBAX SB-C18 column, where the separations of stereoisomers had not been achieved to the results obtained using CHIRALPAK IC column.

Response: The comparison between the content of astaxanthin determined using ZORBAXSB-C18 column has been included in the text as follows.

The content of astaxanthin from different sources was almost the same.

Prior to acceptance, please, perform the experiment with real samples, for example, determine the stereoisomeres in sample originated from salmon for which the color additive astaxanthin was used, in order to verify the suggested method.

Response: This suggestion is very good. That is what we plan to do in the next step. Because the real samples feeding with mixed astaxanthin from different sources can not be obtained in short time, we are sorry that we can not verify the suggested method now.

Overall impression is that the authors successfully employed the mathematical model in solving the problem of quantitative and qualitative analysis of the stereoisomers.

Response: Thank you very much!

Round 2

Reviewer 2 Report

My suggestion were attended by the authors. Then, this paper can be now considered for publication at Molecules. 

Reviewer 3 Report

Manuscript No: molecules 696097

Authors improved their manuscript accordant with the comments made by reviewer.

Still, having no time to check the method if applied in the case of real samples, is something that decreases the significance of the performed investigation and gives the impression that authors did not think through the applicable side of the conducted research. In this form, the research might be considered as preliminary, postponing the publishing the important aspect of possible application.